# Adversarial Blocking Bandits

**Nicholas Bishop**
University of Southampton, UK
nb8g13@soton.ac.uk

**Hau Chan**
University of Nebraska-Lincoln, USA
hchan3@unl.edu

**Debmalya Mandal**
Columbia University, USA
dm3557@columbia.edu

**Long Tran-Thanh**
University of Warwick, UK
long.tran-thanh@warwick.ac.uk

## Abstract

We consider a general adversarial multi-armed blocking bandit setting where each played arm can be *blocked* (unavailable) for some time periods and the reward per arm is given at each time period *adversarially* without obeying any distribution. The setting models scenarios of allocating scarce limited supplies (e.g., arms) where the supplies replenish and can be reused only after certain time periods. We first show that, in the optimization setting, when the blocking durations and rewards are known in advance, finding an optimal policy (e.g., determining which arm per round) that maximises the cumulative reward is strongly NP-hard, eliminating the possibility of a fully polynomial-time approximation scheme (FPTAS) for the problem unless P = NP. To complement our result, we show that a greedy algorithm that plays the best available arm at each round provides an approximation guarantee that depends on the blocking durations and the path variance of the rewards. In the bandit setting, when the blocking durations and rewards are not known, we design two algorithms, `RGA` and `RGA-META`, for the case of bounded duration an path variation. In particular, when the variation budget $B_T$ is known in advance, `RGA` can achieve $\mathcal{O}(\sqrt{T(2\tilde{D} + K)B_T})$ dynamic approximate regret. On the other hand, when $B_T$ is not known, we show that the dynamic approximate regret of `RGA-META` is at most $\mathcal{O}((K + \tilde{D})^{1/4}\tilde{B}^{1/2}T^{3/4})$ where $\tilde{B}$ is the maximal path variation budget within each batch of `RGA-META` (which is provably in order of $o(\sqrt{T})$. We also prove that if either the variation budget or the maximal blocking duration is unbounded, the approximate regret will be at least $\Theta(T)$. We also show that the regret upper bound of `RGA` is tight if the blocking durations are bounded above by an order of $\mathcal{O}(1)$.

## 1   Introduction

This paper investigates the blocking bandit model where pulling an arm results in having that arm blocked for a deterministic number of rounds. For example, consider the classical problem of online task allocation, in which new task requests arrive at each time step, waiting to be assigned to one of many servers [Karthik et al., 2017]. Once a server is allocated to a task, it starts working on it, and becomes unavailable for future tasks until that task is done. If there are no servers available or none is allocated to the task at its arrival, the request will not be served and leave the system forever. A more recent example comes from the domain of expert crowdsourcing (e.g., Upwork, Outsourcely, etc.). In this setting, a job requester can sequentially choose from a pool of workers and allocate a short-term job/project to the worker [Ho and Vaughan, 2012, Tran-Thanh et al., 2014]. The stochastic version of this problem, where the rewards are randomly drawn from a distribution in an

i.i.d. manner, with the constraint that the blocking durations are fixed per arm over time, has been studied in [Basu et al., 2019] and [Basu et al., 2020]. However, in many applications, the stochastic setting is too restrictive and not realistic. For example, in the online task allocation problem, the tasks can be heterogeneous, and both the value and the serving time of the tasks can vary over time in an arbitrary manner. Furthermore, in the expert crowdsourcing setting, the time and quality workers need to deliver the job are unknown in advance, can vary over time, and do not necessarily follow an i.i.d. stochastic process. These examples demonstrate that for many real-world situations, the stochastic blocking bandit model is not an appropriate choice.

To overcome this issue, in this paper we propose the adversarial blocking bandit setting, where both the sequence of rewards and blocking durations per arm can be arbitrary. While the literature of adversarial bandits is enormous, to the best of our knowledge, this is the first attempt to address the effect of blocking in adversarial models. In particular, we are interested in a setting where the rewards are neither sampled i.i.d., nor maliciously chosen in an arbitrary way. Instead, in many real-world systems, the change in the value of rewards is rather slow or smooth over time (e.g., in the online task allocation problem, similar tasks usually arrive in batch, or in the crowdsourcing system, workers may have periods when they perform consistently, and thus, their performance slowly varies over time). To capture this, we assume that there is a path variation budget which controls the change of the rewards over time [1].

## 1.1 Main Contributions

In this paper, apart from the adversarial blocking bandit setting, we also investigate two additional versions of the model: (i) The offline MAXREWARD problem, where all the rewards and blocking durations are known in advance; and (ii) the online version of MAXREWARD, in which we see the corresponding rewards and blocking durations of the arms at each time step *before* we choose an arm to pull. Our main findings can be summarised as follows:

1. We prove that the offline MAXREWARD problem is strongly NP-hard (Theorem 1). Note that this result is stronger than the computational hardness result in Basu et al. [2019], which depends on the correctness of the randomised exponential time hypothesis.

2. We devise a provable approximation ratio for a simple online greedy algorithm, `Greedy-BAA`, for the online MAXREWARD problem (Theorem 2). Our approximation ratio, when applied to the stochastic blocking bandit model with fixed blocking durations, is slightly weaker than that of Basu et al. [2019]. However, it is more generic, as it can be applied to any arbitrary sequence of rewards and blocking durations.

3. For the bandit setting, we consider the case when both the maximal blocking duration and the path variance are bounded, and propose two bandit algorithms:

- We design `RGA` for the case of known path variation budget $B_T$. In particular, we show that `RGA` can provably achieve $\mathcal{O}\left(\sqrt{T(2\tilde{D} + K)B_T}\right)$ regret, where $T$ is the time horizon, $K$ is the number of arms, $\tilde{D}$ is the maximum blocking duration, and the regret is computed against the performance of `Greedy-BAA` (Theorem 3).

- For the case of unknown path variation budget $B_T$, we propose `RGA-META` that uses Exp3 as a meta-bandit algorithm to learn an appropriate path variation budget and runs `RGA` with it. We prove that `RGA-META` achieves $\mathcal{O}((K + \tilde{D})^{1/4}\tilde{B}^{1/2}T^{3/4})$ regret bound where $\tilde{B}$ is the maximal path variance within a single batch of the algorithm, and is in order of $\mathcal{O}(\sqrt{T})$ in the worst case (Theorem 4).

4. Finally, we also discuss a number of regret lower bound results. In particular, we show that if either $B_T$ or $\tilde{D}$ is in $\Theta(T)$ (or unbounded), then the regret is at least $\Theta(T)$ (Claims 1 and 2). We also discuss that if $\tilde{D} \in \mathcal{O}(1)$, then there is a matching lower bound for the regret of `RGA` (Section 5).

## 1.2 Related Work

**Stochastic Blocking Bandits.** The most relevant work to our setting is the stochastic blocking bandit model. As mentioned before, Basu et al. [2019] introduce and study this model where the reward per each time period is generated from a stochastic distribution with mean $\mu_k$ reward for each arm $k$ and the blocking duration is fixed across all time period for each arm $k$ (e.g., $D_t^k = D^k$ for all $t$ and $k$). In the optimization setting where the mean rewards and blocking durations are known, they consider a simpler version of the MAXREWARD problem for their setting and show that the problem is as hard as the PINWHEEL Scheduling on dense instances [Jacobs and Longo, 2014] and provide that a simple greedy algorithm (see Algorithm 1) achieves an approximation ratio of $(1 - 1/e - O(1/T))$ where $T$ is total time period. In the bandit setting, they provide lower and upper regret bounds that depend on the number of arms, mean rewards, and $\log(T)$. A very recent work [Basu et al., 2020] extends the stochastic blocking bandit to a contextual setting where a context is sampled according to a distribution each time period and the reward per arm is drawn from a distribution with the mean depending on the pulled arm and the given context. Similar to the work of Basu et al. [2019], Basu et al. [2020] derive an online algorithm with an approximation ratio that depends on the maximum blocking durations and provide upper and lower $\alpha$-regret bounds of $O(\log T)$ and $\Omega(\log T)$, respectively. However, the results from this models cannot be directly applied to the adversarial setting due to the differences between the stochastic and adversarial reward generation schemes.

**Budgeted and Knapsack Bandits.** Since the underlying offline optimisation problem of our setting, MAXREWARD, can also be casted as an instance of the multiple-choice multidimensional knapsack problem, it is also worth mentioning the line of work in the bandit literature that solve online knapsack problems with bandit feedback. In these models, the pull of an arm requires the consumption of resources in $d \geq 1$ dimensions. The resource per arm is given either stochastic or adversarially in each time period and a (non replenishable) total budget $B = (B_1, ..., B_d)$ is available at the initial time period. The one-dimensional stochastic version of this setting is first studied in Tran-Thanh et al. [2010, 2012], Ding et al. [2013] under the name budgeted bandits, and is later extended to multiple dimensions (a.k.a. bandits with knapsack) by Badanidiyuru et al. [2013], Agrawal and Devanur [2014], Badanidiyuru et al. [2014]. More recently, Rangi et al. [2019] and Immorlica et al. [2019] initiate the study of adversarial knapsack bandits. Rangi et al. [2019] consider the $d = 1$ setting with a regret benchmark that is measured based on the best fixed-arm's reward to cost ratio. Under such a regret benchmark, they show that sub-linear regret (with respect to $B$ and $k$) is possible in both the stochastic and adversarial settings. Immorlica et al. [2019] consider the $d \geq 1$ setting with a regret benchmark that is defined to be the ratio of the expected reward of the best fixed distribution over arms and the policy's expected reward. show that the ratio is at least $\Omega(\log T)$. However, none of the techniques developed in these work can be applied to our setting, due to the following reason: The results in the knapsack bandit models typically assume that the pulling costs are bounded above by a constant, and the budget is significantly larger than this constant to allow sufficient exploration. In contrast, when MAXREWARD is conversed into a knapsack model, many of its dimensions will have a budget of $1$, and the corresponding pulling cost for that dimension is also $1$ (due to the blocking condition).

**Other Settings with Arm Availability Constraints.** Other bandit models with arm availability cosntrainsts include the mortal bandits [Chakrabarti et al., 2009], sleeping bandits [Kleinberg et al., 2010, Kale et al., 2016], bandits with stochastic action sets [Neu and Valko, 2014], and combinatorial semi-bandits [Neu and Bartók, 2016]. We refer readers to [Basu et al., 2019] for a discussion of these models, including the relevance of the blocking bandit setting to online Markov decision processes.

**Connection to the scheduling literature.** Notice that there is a strong connection between MAXRE-WARD and the interval scheduling problems. In particular, the MAXREWARD problem belongs to the class of fixed interval scheduling problems with arbitrary weight values, no preemption, and machine dependent processing time (see e.g., Kolen et al. [2007] for a comprehensive survey). This is one of the most general, and thus, hardest versions of the fixed interval scheduling literature (see, e.g., Kovalyov et al. [2007] for more details). In particular, MAXREWARD is a special case of this setting where for each task, the starting point of the feasible processing interval is equal to the arrival time. Note that to date, provable performance guarantees for fixed interval scheduling problems with arbitrary weight values only exist in offline, online but preemptive, or settings with some special uniformity assumptions (e.g., [Erlebach and Spieksma, 2000, Miyazawa and Erlebach, 2004, Bender et al., 2017, Yu and Jacobson, 2020]). Therefore, to our best knowledge, Theorem 2 in our paper is

the first result which provides provable approximation ratio for a deterministic algorithm in an online non-preemptive setting. Note that with some modifications, our proof can also be extended to the general online non-preemptive setting, i.e., online interval scheduling with arbitrary weight values, no preemption, and machine dependent processing time.

## 2   Preliminaries

**Adversarial blocking bandit.** In this paper we consider the following bandit setting. Let $\mathcal{K} = \{1, \ldots, K\}$ be the set of $K$ arms. Let $\mathcal{T} = \{1, \ldots, T\}$ denote a sequence of $T$ time steps, or decision points faced by a decision maker. At every time step $t \in \mathcal{T}$, the decision maker may pull one of the $K$ arms. When pulling an arm $k \in \mathcal{K}$ at time step $t \in \mathcal{T}$, the reward $X_t^k \in [0, 1]$ is obtained. In addition, the pulled arm $k$ is deterministically blocked and cannot be pulled for the next $(D_t^k - 1)$ time steps for some integer blocking duration $D_t^k \in \mathbb{Z}^+$. We also use the notation $\emptyset$ to denote the action of not pulling an arm. In which case, $X_t^\emptyset = 0$ and $D_t^\emptyset = 1$ for each time step $t$.

We denote by $X^k$ the sequence of rewards over $T$ time steps associated with an arm $k \in \mathcal{K}$ such that $X^k = \{X_t^k\}_{t=1}^T$. In addition, we denote by $X$ the sequence of vectors of all $K$ rewards such that $X = \{X^k\}_{k=1}^K$. Similarly, we denote by $D^k = \{D_t^k\}_{t=1}^T$ the sequence of blocking durations over $T$ time steps associated with an arm $k$ and denote by $D = \{D^k\}_{k=1}^K$ the sequence of vectors of all $K$ blocking duration vectors.

In our model, the rewards and blocking durations of each arm can change an arbitrary number of times. We let $\tilde{D}$ ($\underset{\sim}{D}$) be the *maximal blocking duration* (*minimal blocking duration*) which is the upper bound (lower bound) of the largest (smallest) possible blocking duration. We denote by $\mathcal{D} = \{1, \ldots, \tilde{D}\}^{K \times T}$ the set of all blocking duration vector sequences which are upper bounded by $\tilde{D}$. Note that $D$ is defined with respect to minimal blocking duration $\underset{\sim}{D} = 1$. It is sometime be useful to define $D$ for arbitrarily lower bound $\underset{\sim}{D}$.

**Bounded path variation.** Motivated by and adapted from a recent line of work in the bandit literature (e.g., [Besbes et al., 2014]), we assume that there is a *path variation budget* on the sequence of the rewards. In particular, the definition of path variation on the sequence of the rewards is defined to be

$$\sum_{t=1}^{T-1} \sum_{k=1}^{K} \left| X_{t+1}^k - X_t^k \right|.$$

We refer to $B_T$ as the path variation budget over $\mathcal{T}$. We define the corresponding temporal uncertainty set as the set of reward vector sequences which satisfy the variation budget over the set of time steps $\{1, \ldots, T\}$:

$$\mathcal{B} = \left\{ X \in [0, 1]^{K \times T} \ : \ \sum_{t=1}^{T-1} \sum_{k=1}^{K} \left| X_t^k - X_{t+1}^k \right| \le B_T \right\}$$

Note that by setting $B_T = KT$ we can recover the standard unbounded version of our bandit model (as all the rewards are from $[0, 1]$). Note that our analysis also works for other variation budgets such as the maximum variation [Besbes et al., 2014] or the number of changes budgets [Auer et al., 2019]. See Section 5 for a more detailed discussion.

**Arm pulling policy.** Let $U$ be a random variable defined over a probability space $(\mathbb{U}, \mathcal{U}, \mathbf{P}_u)$ Let $\pi_1 : \mathbb{U} \to \mathcal{K}$ and $\pi_t : [0, 1]^{t-1} \times \{1, \ldots, \tilde{D}\}^{t-1} \times \mathbb{U} \to \mathcal{K}$ for $t = 2, 3, \ldots$ be measurable functions With some abuse of notation we denote by $\pi_t \in \mathcal{K}$ the arm chosen at time $t$, that is given by

$$\pi_t = \begin{cases} \pi_1(U) & t = 1 \\ \pi_t(X_{t-1}^\pi, \ldots, X_1^\pi, D_{t-1}^\pi, \ldots, D_1^\pi, U) & t = 2, 3, \ldots \end{cases}$$

Here $X_t^\pi$ (resp. $D_t^\pi$) denotes the reward (resp. blocking duration) observed by the policy $\pi$ at time $t$ The mappings $\{\pi_t \ : \ t = 1, \ldots, T\}$ together with the distribution $\mathbf{P}_u$ define the class of policies We define the class $\mathcal{P}$ of admissible policies to be those, at every time step, which choose an action which is not blocked. That is,

$$\mathcal{P} = \left\{ (\pi_1, \ldots, \pi_T) \ : \ \pi_t \notin \{\pi_j : j + D_j^{\pi_j} - 1 \ge t, \ \forall j \le t - 1\}, \ \forall t \in \{1, \ldots, T\}, X \in \mathcal{B}, D \in \mathcal{D} \right\}.$$

---
**Algorithm 1:** `Greedy-BAA`
---
**Input** : $T, K, \{X_t^k\}_{k \in \mathcal{K}, t \in \mathcal{T}}, \{D_t^k\}_{k \in \mathcal{K}, t \in \mathcal{T}}$ - An instance of the MAXREWARD Problem
**Output** : $\pi^+ = (\pi_1^+, \pi_2^+, ..., \pi_T^+) \in \mathcal{P}$ - A greedy solution to the MAXREWARD Problem
**1** $\pi^+ = (\emptyset, ..., \emptyset)$;
**2** **for** $j \leftarrow 1$ **to** $T$ **do**
**3** $\quad$ Select $\pi_j^+ \in \arg\max_{k_j \in A_j(\pi_1^+, ..., \pi_{j-1}^+) \cup \emptyset} X_j^{k_j}$ $\quad$ # See the preliminary section for definitions
**4** **end**
**5** **return** $\pi^+$
---

In addition, let $A_t(\pi_1, \dots, \pi_{t-1}) = \mathcal{K} \setminus \{\pi_j : j + D_j^{\pi_j} - 1 \geq t, \; \forall j \leq t-1\}$ denote the set of available arms at time step $t$ (we will also use $A_t$ for the sake of brevity).

**Objective.** The cumulative reward of a policy $\pi \in \mathcal{P}$ is defined to be $r(\pi) = \sum_{t=1}^{T} X_t^\pi$ where $X_t^\pi$ is the reward obtained by policy $\pi$ at time step $t$. Our objective is to find $\pi^* \in \mathcal{P}$ such that $\pi^* \in \arg\max_{\pi \in \mathcal{P}} \mathbb{E}^\pi[r(\pi)]$, where the expectation is over all possible randomisation coming from policy $\pi$.

**Feedback.** The difficulty of the optimisation problem depends on the information (or the feedback) we have about the rewards and blocking durations of the arms. In this paper, we consider three feedback models in increasing order of difficulty. In the simplest setting, we know the value of all $X_t^k$ and $D_t^k$ in advance. We refer to this setting as the (offline) MAXREWARD optimization problem. In the online version of MAXREWARD, we assume that $X_t^k$ and $D_t^k$ are not known in advance, but at each time step $t$, the value of $X_t^k$ and $D_t^k$ for all $k$ at that particular time step $t$ is revealed before we choose any arm to pull. Finally, in the (classical) bandit setting, we assume that only the reward and blocking duration of the chosen arms are revealed after that arm is pulled[2]. We will refer to third model as the *adversarial blocking bandit problem*.

## 3 The Offline and Online MAXREWARD Problems

We start with the analysis of the offline and online MAXREWARD problems. As a slight preview of the next subsections, computing an optimal solution of the offline MAXREWARD problem is strongly NP-hard even with bounded variation budget. Such result eliminates the possibility of a fully polynomial-time approximation scheme (FPTAS) for the problem unless P = NP. In addition, for the online MAXREWARD problem, we design an online greedy algorithm with provable approximation guarantee.

### 3.1 The Computational Complexity of the Offline MAXREWARD Problem

To show that the MAXREWARD problem is strongly NP-hard, we reduce from the Boolean satisfiability problem with three literals per clause (3-SAT), which is known to be strongly NP-complete [Garey and Johnson, 1979]. In a 3-SAT instance, we are given $m$ variables and $n$ clauses. Each clause consists of three literals, and each literal is either a variable or the negation of the variable. The problem is to determine if there is a boolean true/false assignment to each variable so that the given 3-SAT instance is true (i.e., each clause contains at least one true literal).

**Theorem 1.** *Computing an optimal solution for the MAXREWARD problem is strongly NP-hard. The hardness result holds even when the path variation is bounded.*

### 3.2 Online MAXREWARD Problem with Bounded Variation Budget

In this section, we consider the online version of MAXREWARD. We devise a simple online greedy algorithm, Greedy Best Available Arm (`Greedy-BAA`), in which, at each time step, the algorithm plays an available arm with the highest reward. Algorithm 1 provides a detail description of `Greedy-BAA`.

Below, we show that `Greedy-BAA` provides an approximation guarantee to the offline MAXREWARD problem that depends on the blocking durations and the variation budget.

**Theorem 2.** *Let $k^* = \arg\max_k \frac{D^k_{\max}}{D^k_{\min}}$ denote the arm with the highest max-min blocking duration ratio. Let $\pi^+$ denote the solution returned by `Greedy-BAA`, and $\pi^*$ denote an optimal solution of the offline MAXREWARD problem, respectively. We state that:*

$$\left(1 + \frac{D^{k^*}_{\max}}{D^{k^*}_{\min}}\right) r(\pi^+) + \frac{D^{k^*}_{\max}}{D^{k^*}_{\min}} B_T \geq r(\pi^*),$$

*That is, `Greedy-BAA` has an approximation ratio of $\left(1 + \frac{D^{k^*}_{\max}}{D^{k^*}_{\min}}\right)^{-1} \left(1 - \frac{D^{k^*}_{\max} B_T}{D^{k^*}_{\min} r(\pi^*)}\right).$*

Note that as $D^{k^*}_{\min} \geq \underset{\sim}{D}$ and $D^{k^*}_{\max} \leq \tilde{D}$, the approximation ratio above can be further bounded above by $\left(1 + \frac{\tilde{D}}{\underset{\sim}{D}}\right)^{-1} \left(1 - \frac{\tilde{D} B_T}{\underset{\sim}{D} r(\pi^*)}\right).$

**Comparison to the result of** Basu et al. [2019]**.** We note that Basu et al. [2019] has studied the MAXREWARD problem with path variation budget $B_T = 0$ (i.e., the reward values are fixed over time) and homogeneous blocking durations per arm (i.e., when the blocking duration per arm do not change over time). In that case, our proof provides an approximation ratio of $1/2$ whereas Basu et al. [2019] provides an approximation ratio of $O(1 - 1/e - O(1/T))$. Their technique uses a much complicated LP-bounding technique/proof that does not directly generalize to the case of $B_T > 0$ with varying blocking durations. On the other hand, our approximation ratio result holds for the general case. For example, if $B_T$ grows slower than $r(\pi^+)$ with $T$, our algorithm guarantees an approximation ratio of $(1 + 2\frac{\tilde{D}}{\underset{\sim}{D}})^{-1}$.

## 4 The Adversarial Blocking Bandit Problem

Given the investigation of the (offline and online) MAXREWARD problems in the previous section, we now turn to the main focus of our paper, namely the online MAXREWARD problem with bandit feedback, a.k.a the adversarial blocking bandit problem. While the regret analyses are typically done by benchmarking against the best fixed policy in hindsight, we can easily show that in our setting, this benchmark would perform arbitrarily poorly, compared to the offline optimal solution. Therefore, instead of following the standard regret analysis, we are interested in comparing the performance of the designed algorithms to that of the offline optimal solution. Therefore, we will use the following regret definition:

**Dynamic approximate regret.** We compare the performance of a policy with respect to the dynamic oracle algorithm that returns the offline optimal solution of MAXREWARD .We define the $\alpha$-regret under a policy $\pi \in \mathcal{P}$ as the worst case difference between an (offline) $\alpha$-optimal sequence of actions and the expected performance under policy $\pi$. More precisely, let $\pi^*$ denote the arm pulling policy of that dynamic oracle algorithm. The $\alpha$-regret of a policy $\pi \in \mathcal{P}$ against $\pi^*$ is defined to be

$$\mathcal{R}^\alpha_\pi(B_T, \tilde{D}, T) = \alpha r(\pi^*) - \mathbb{E}[r(\pi)]$$

where the expectation is over all the possible randomisation of $\pi$. Note that this regret notion is stronger than the regret against the best fixed policy in hindsight, as it is easy to show that the best fixed policy can perform arbitrarily badly, compared to $\pi^*$.

### 4.1 Blocking Bandit with Known Path Variation Budget

We now turn to describe our new bandit algorithm, `RGA`, designed for the adversarial blocking bandit problem. This algorithm can be described as follows:

1. We split the time horizon $\mathcal{T}$ into batches $\mathcal{T}_1, \ldots, \mathcal{T}_m$ of size $\Delta_T$ each (except possibly the last batch):

$$\mathcal{T}_j = \{t \in \{1, \ldots, \Delta_T\} \; : \; (j-1)\Delta_T + t \leq \min\{j\Delta_T, T\}\}, \quad \text{for all } j = 1, \ldots, m$$

where $m = \left\lceil \frac{T}{\Delta_T} \right\rceil$ is the number of batches.

---

**Algorithm 2:** Repeating Greedy Algorithm (`RGA`)

**Input:** $\Delta_T$.

**1** **while** $1 \leq j \leq \left\lceil \frac{T}{\Delta_T} \right\rceil$ **do**
**2** $\quad$ Set $\tau = 1$
**3** $\quad$ **while** $\tau \leq \Delta_T$ **do**
**4** $\quad\quad$ **if** *($1 \leq \tau \leq K$)* **then**
**5** $\quad\quad\quad$ Pull arm $k = \tau \mod K + 1$
**6** $\quad\quad\quad$ Receive reward and blocking duration $(X_\tau^k, D_\tau^k)$
**7** $\quad\quad\quad$ Set $\hat{X}_t^k = X_\tau^k$ for all $t \in [1, \Delta_T]$.
**8** $\quad\quad$ **if** *($K + 1 \leq \tau \leq \tilde{D} + K$)* **then**
**9** $\quad\quad\quad$ Pull no arms
**10** $\quad\quad$ **if** *($\tilde{D} + K + 1 \leq \tau \leq \Delta_T - \tilde{D}$)* **then**
**11** $\quad\quad\quad$ Pick arms according to `GREEDY-BAA`$(\Delta_T - 2\tilde{D} - K, K, \hat{X}^1, \ldots, \hat{X}^K, D^1, \ldots, D^K)$
**12** $\quad\quad$ **if** *($\Delta_T - \tilde{D} + 1 \leq \tau \leq \Delta_T$)* **then**
**13** $\quad\quad\quad$ Pull no arms
**14** $\quad\quad$ $\tau \leftarrow \tau + 1$
**15** $\quad$ $j \leftarrow j + 1$

---

2. Within each batch we spend the first $K$ rounds pulling each arm. Without loss of generality, we shall assume that arm $k$ is pulled on round $k$. After this we spend the next $\tilde{D}$ rounds pulling no arms. This ensures that all arms will be available when we next pull an arm.

3. Then, up until the final $\tilde{D}$ rounds we play `Greedy-BAA` using the rewards observed in the first $K$ rounds as the fixed rewards for each arm.

4. In the final $\tilde{D}$ rounds of each batch, we again pull no arms. This ensures that all of the arms are available at the beginning of the next batch.

**Theorem 3.** *Suppose that the variation budget $B_T$ is known in advance and maximal duration $\tilde{D} \geq 1$ such that $\tilde{D}B_T \in o(T)$. The $\alpha$-regret of `RGA`, where $\alpha = \frac{D}{\tilde{D}+D}$, is at most $\mathcal{O}\left( \sqrt{T(2\tilde{D} + K)B_T} \right)$ when the parameter when $\Delta_T$ is set to $\left\lceil \sqrt{\frac{(T+1)(2\tilde{D}+K)}{2B_T}} \right\rceil$.*

Note that this bound is sub-linear in $T$ if $\tilde{D}B_T = o(T)$ (e.g., $\tilde{D}$ is bounded above by a constant and $B_T \in o(T)$). It is also worth noting that while $\alpha = \frac{1}{1+\tilde{D}}$ might imply that `RGA` can perform better than the worst-case performance of `Greedy-BAA`, with $B_T \in o(T)$ it is not the case (see Section **??** in the appendix for more details).

### 4.2 Blocking Bandit with Unknown Path Variation Budget

Note that `RGA` requires knowledge of $B_T$ in order to properly set $\Delta_T$. To resolve this issue we propose `META-RGA`, a meta-bandit algorithm, where each arm corresponds to an instance of the `RGA` algorithm whose $\Delta_T$ parameter tuned for a different variation budget. The time horizon $\mathcal{T}$ is broken into meta-blocks of length $H$. At the start of each meta-block an arm (i.e., an instance of `RGA` with its corresponding budget) is selected according to the well known Exp3 algorithm [Auer et al., 2002]. The `RGA` is then played for the next $H$ time steps with optimally tuned restarts (see Theorem 3 for more details). At the end of the meta-block, the Exp3 observes a reward corresponding to the total reward accumulated by the chosen `RGA` in this meta-block. The intuition of this idea is that the meta-bandit will learn which budget will be the best upper bound for `RGA`.

In what follows, we shall denote the set of arms available to the Exp3 algorithm by $\mathcal{J}$, and denote the corresponding set of variation budgets by $\mathcal{J}_B$. The `META-RGA` algorithm uses $\lceil \log_2(KT) \rceil + 1$ meta-arms with budgets $\mathcal{J}_B = \{2^0, 2^1, \ldots, 2^{\lceil \log_2(KT) \rceil}\}$. That is, the budget values are powers of 2 up to the smallest 2-power, which is still larger than $KT$, which is the ultimate upper bound of the

**Algorithm 3:** Meta Repeating Greedy Algorithm (`META-RGA`)

**Input:** $T, K, \gamma \in (0, 1]$, batch length $H$.

1 **Initialize**: $|\mathcal{J}| = \lceil \log_2(KT) \rceil + 1$, $\mathcal{J}_B = \{2^0, 2^1, \ldots, 2^{\lceil \log_2(KT) \rceil}\}$, $w_i(1) = 1$ for $i = 1, \ldots, |\mathcal{J}|$.

2 **for** $\tau = 1, \ldots, \lceil \frac{T}{H} \rceil$ **do**

3     Set

$$p_i(\tau) = (1 - \gamma)\frac{w_i(\tau)}{\sum_{j=1}^{|\mathcal{J}|} w_j(\tau)} + \frac{\gamma}{|\mathcal{J}|} \quad i = 1, \ldots, |\mathcal{J}|$$

4     Draw $i_\tau$ randomly according to the probabilities $p_1(\tau), \ldots, p_{|\mathcal{J}|}(\tau)$

5     Run `RGA` in batch $\tau$ with budget $\mathcal{J}_B[i_\tau] = 2^{i_\tau - 1}$ and optimally tuned restarts

6     Receive reward $x_{i_t}(\tau) \in [0, H]$ at the end of the batch

7     **for** $j = 1, \ldots, |\mathcal{J}|$ **do**

$$\hat{x}_j(\tau) = \begin{cases} \frac{x_j(\tau)}{p_j(\tau)} & \text{if } j = i_\tau \\ 0 & \text{otherwise} \end{cases}$$

$$w_j(\tau + 1) = w_j(\tau)\exp(\gamma\hat{x}_j(\tau)/(H|\mathcal{J}|))$$

path variation budget (as $B_T \leq KT$). In addition, let $B_i$ denote the total path variance within batch $i$, and $\tilde{B} = \max_i B_i$. We state the following:

**Theorem 4.** *Suppose that the variation budget $B_T$ is unknown in advance to us. In addition, suppose that the maximal blocking duration $\tilde{D} \geq 1$ such that $\tilde{D}B_T \in \mathrm{o}(T)$. The $\alpha$-regret of `RGA-META`, where $\alpha = \frac{1}{1+\tilde{D}}$, is at most*

$$\mathcal{O}\left(\tilde{B}^{1/2}T^{3/4}(2\tilde{D} + K)^{1/4}\ln(KT)^{1/4}\ln(\ln(KT))^{1/4}\right)$$

*when the parameters of `RGA-META` are set as follows:*

$$H = \sqrt{\frac{T(2\tilde{D} + K)}{\ln(KT)\ln(\ln(KT))}}, \quad \gamma = \min\left\{1, \sqrt{\frac{\ln(KT)\ln(\ln(KT))}{(e - 1)T}}\right\}.$$

Note that since $\tilde{B} \leq HK$ by definition (the maximum path variance within a batch is at most $HK$), by setting $H = \sqrt{\frac{T(2\tilde{D} + K)}{\ln(KT)\ln(\ln(KT))}}$ we always get sub-linear regret in $T$ if $\tilde{D} \in \mathcal{O}(1)$ (i.e., is bounded above by a constant). Otherwise we need to have $\tilde{B}^2\tilde{D} \in \mathrm{o}(T)$. Furthermore, when $\tilde{B}$ is small, our regret bound tends to $\mathcal{O}(T^{3/4})$. Thus, it is still an open question whether we can get a tighter upper bound (e.g., $\mathcal{O}(\sqrt{T})$) for this case (i.e., when the variation budget is unknown).

## 5 Discussions

In this section we will provide some intuitions why we set $B_T$ and $\tilde{D}$ to be small in the previous sections. In particular, we show that if either the variation budget or the maximum blocking duration is large, the lower bound of the $\alpha$-regret is $\Theta(T)$. We also discuss a potential lower bound for the $\alpha$-regret of the adversarial blocking bandit problem in the case of $B_T \in \mathrm{o}(KT)$ and $\tilde{D} \in \mathcal{O}(1)$. Finally, we will also discuss how our results change if we use other types of variation budgets.

**Large variation budget.** Consider the case when $B_T \in \Theta(T)$. Theorem 3 indicates that the upper bound of the $\alpha$-regret is $\Theta(T)$ where $\alpha = \frac{1}{1+\tilde{D}}$ as defined in Theorem 3. Indeed, we show that this is the best possible we can achieve:

**Claim 1.** *For any $T > 0$ and $B_T \in \Theta(KT)$, there exists a sequence of rewards and blocking durations $X$ and $D$ such that $\mathcal{R}_\pi^\alpha(B_T, \tilde{D}, T) = \Theta(T)$ for that particular $(X, D)$.*

**Large blocking durations.** If $\tilde{D} \in \Theta(T)$ and $\alpha$ is the approximation ratio of `Greedy-BAA`:

**Claim 2.** *For any $T > 0$ and $\tilde{D} \in \Theta(T)$, there exists a sequence of rewards and blocking durations $X$ and $D$ such that $\mathcal{R}_\pi^\alpha(B_T, \tilde{D}, T) = \Theta(T)$ for that particular $(X, D)$.*

Note that our regret bounds only make sense if $\tilde{D}B_T \in o(T)$. Thus, it is still an open question whether we can achieve sub-linear $\alpha$-regret bounds in $T$ if both $B_T, \tilde{D} \in o(T)$ but $\tilde{D}B_T \in \Omega(T)$.

**Almost matching regret lower bound for RGA.** Consider the case when $\tilde{D} = \mathcal{O}(1)$. This implies that the $\alpha$-regret bound of RGA is reduced to $\mathcal{O}(\sqrt{KTB_T})$. This in fact matches the known lower bounds of the 1-regret for the case of $\tilde{D} = 1$ (i.e., no blocking) from the literature [Auer et al., 2019]. In particular, with $\tilde{D} = 1$, the Greedy-BAA algorithm becomes optimal (see, e.g., Section 4.3 of Basu et al. [2019] for the discussion of this), and thus, the $\alpha$-regret notion becomes 1-regret. Therefore, if there exists an algorithm which could achive an $\alpha$-regret better than $\mathcal{O}(\sqrt{KTB_T})$ in our setting, then it would be able to achieve $\mathcal{O}(\sqrt{KTB_T})$ 1-regret for the standard (i.e., non-blocking) adversarial bandit as well.

It is also worth noting that when $\tilde{D}$ is not bounded above by a constant, or the variation budget $B_T$ is not known in advance, it is still not known what the regret lower bound would be.

**Other variation budget definitions.** There are a number of different variation budget definitions in the literature [Besbes et al., 2014, Wei and Luo, 2018, Auer et al., 2019]. It is worth noting that our analysis works in a similar way for the maximum variation budget $B_T^{\max}$ and number of changes budget $L_T$, which can be defined as follows:

$$B_T^{\max} = \sum_{t,t+1 \in \mathcal{T}} \max_{k \in \mathcal{K}} |X_{t+1}^k - X_t^k|, \quad L_T = \#\{t : 1 \le t \le T - 1, \exists k : X_t^k \ne X_{t+1}^k\}$$

If we use these variation budgets instead, the regret in Theorem 3 will be modified to $\mathcal{O}\left(\sqrt{(2\tilde{D} + K)TB_T^{\max}}\right)$ and $\mathcal{O}\left(\sqrt{(2\tilde{D} + K)TL_T}\right)$, respectively. Furthermore, the approximation ratio of Greedy-BAA will also change. In particular, it becomes $\left[\left(1 + \tilde{D}\right) + \tilde{D}KB_T^{\max}/r(\pi^+)\right]^{-1}$ and $\left[\left(1 + \tilde{D}\right) + \tilde{D}KL_T/r(\pi^+)\right]^{-1}$. We refer the reader to Section **??** in the appendix for a more detailed discussion. It remains as future work to derive regret bounds for the other variation budgets.

## Broader Impact

The paper examines a novel multi-armed bandit problem in which the decision-making agent aims to receive as many (cumulative) rewards as possible over a finite period subject to constraints. Our focus and results are largely theoretical. In particular, our contributions advance our understanding of multi-armed bandit models and its theoretical limitations and benefit the general (theoretical) machine learning community, specifically the multi-armed bandit and online learning communities. In addition, we do not expect that our theoretical findings can be directly used in more applied domains.

## Acknowledgments and Disclosure of Funding

Nicholas Bishop was supported by the UK Engineering and Physical Sciences Research Council (EPSRC) Doctoral Training Partnership grant. Debmalya Mandal was supported by a Columbia Data Science Institute Post-Doctoral Fellowship.

## Footnotes

[1]We will show in Section 5 that bounded variation budgets are necessary to achieve sub-linear regrets.

[2]In this paper, due to space limits, we do not deal with the full information feedback model, in which the reward and blocking duration values of all the arms are revealed at each time step after the pull.

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
