[Supplementary Material]

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

_{\max}^{k^*}}{D_{\min}^{k^*}}\right)^{-1}\left(1 - \frac{D_{\max}^{k^*} B_T}{D_{\min}^{k^*} r(\pi^*)}\right)$.*

Note that as $D_{\min}^{k^*} \geq \underset{\sim}{D}$ and $D_{\max}^{k^*} \leq \tilde{D}$, the approximation ratio above can be further bounded above by $\left(1 + \frac{\tilde{D}}{\underset{\sim}{D}}\right)^{-1}\left(1 - \frac{\tilde{D} B_T}{\underset{\sim}{D} r(\pi^*)}\right)$.

**Comparison to the result of** Basu et al. [2019]. We note that Basu et al. [2019] has studied the MAXREWARD problem with path variation budget $B_T = 0$ (i.e., the reward values are fixed over time) and homogeneous blocking durations per arm (i.e., when the blocking duration per arm do not change over time). In that case, our proof provides an approximation ratio of $1/2$ whereas Basu et al. [2019] provides an approximation ratio of $O(1 - 1/e - O(1/T))$. Their technique uses a much complicated LP-bounding technique/proof that does not directly generalize to the case of $B_T > 0$ with varying blocking durations. On the other hand, our approximation ratio result holds for the general case. For example, if $B_T$ grows slower than $r(\pi^+)$ with $T$, our algorithm guarantees an approximation ratio of $(1 + 2\frac{\tilde{D}}{\underset{\sim}{D}})^{-1}$.

# 4 The Adversarial Blocking Bandit Problem

Given the investigation of the (offline and online) MAXREWARD problems in the previous section, we now turn to the main focus of our paper, namely the online MAXREWARD problem with bandit feedback, a.k.a the adversarial blocking bandit problem. While the regret analyses are typically done by benchmarking against the best fixed policy in hindsight, we can easily show that in our setting, this benchmark would perform arbitrarily poorly, compared to the offline optimal solution. Therefore, instead of following the standard regret analysis, we are interested in comparing the performance of the designed algorithms to that of the offline optimal solution. Therefore, we will use the following regret definition:

**Dynamic approximate regret.** We compare the performance of a policy with respect to the dynamic oracle algorithm that returns the offline optimal solution of MAXREWARD .We define the $\alpha$-regret under a policy $\pi \in \mathcal{P}$ as the worst case difference between an (offline) $\alpha$-optimal sequence of actions and the expected performance under policy $\pi$. More precisely, let $\pi^*$ denote the arm pulling policy of that dynamic oracle algorithm. The $\alpha$-regret of a policy $\pi \in \mathcal{P}$ against $\pi^*$ is defined to be

$$\mathcal{R}_\pi^\alpha(B_T, \tilde{D}, T) = \alpha r(\pi^*) - \mathbb{E}[r(\pi)]$$

where the expectation is over all the possible randomisation of $\pi$. Note that this regret notion is stronger than the regret against the best fixed policy in hindsight, as it is easy to show that the best fixed policy can perform arbitrarily badly, compared to $\pi^*$.

## 4.1 Blocking Bandit with Known Path Variation Budget

We now turn to describe our new bandit algorithm, `RGA`, designed for the adversarial blocking bandit problem. This algorithm can be described as follows:

1. We split the time horizon $\mathcal{T}$ into batches $\mathcal{T}_1, \ldots, \mathcal{T}_m$ of size $\Delta_T$ each (except possibly the last batch):

$$\mathcal{T}_j = \{t \in \{1, \ldots, \Delta_T\} \; : \; (j-1)\Delta_T + t \leq \min\{j\Delta_T, T\}\}, \quad \text{for all } j = 1, \ldots, m$$

where $m = \left\lceil \frac{T}{\Delta_T} \right\rceil$ is the number of batches.

---

**Algorithm 2:** Repeating Greedy Algorithm (`RGA`)

**Input:** $\Delta_T$.

1 **while** $1 \leq j \leq \left\lceil \frac{T}{\Delta_T} \right\rceil$ **do**
2      Set $\tau = 1$
3      **while** $\tau \leq \Delta_T$ **do**
4          **if** *($1 \leq \tau \leq K$)* **then**
5              Pull arm $k = \tau \mod K + 1$

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

# A  Proofs for Theorems from Section 3

*Proof of Theorem 1.*  Given a 3-SAT instance of $m$ variables, $v_1, ..., v_m$, and $n$ clauses, $C_1, ..., C_n$, we construct an instance of the MAXREWARD (decision) problem as follows.

- For each variable $v_j$, we create two arms $k_j$ and $\bar{k}_j$.

- For each variable $v_j$, we set $X_j^{k_j} = X_j^{\bar{k}_j} = 1$ and $D_j^{i_j} = D_j^{\bar{i}_j} = T_0$ for some $T_0 > 0$ (all other rewards and blocking durations are set to zero and one, respectively, by default at this point).

- For each clause $C_l$ and each literal $l(v)$ in $C_l$, we set $X_{m+l}^{l(v)} = 1$. Note that all other rewards and blocking durations are also zero and one, respectively, except those from the above.

- We let $T_0 = m + n$, $V = n + m$ and $B_{T_0} = O(T_0 m)$.

Clearly, the constructed instance's parameters are polynomial bounded. Given the MAXREWARD (decision) problem instance, our goal is to find $\pi^* = (\pi_1^*, ..., \pi_T^*) \in \mathcal{P}$ such that $r(\pi^*) \geq V$.

**Claim 3.** *There a solution to the MAXREWARD problem if only if there is a solution to 3-SAT.*

*Proof of Claim 3.* *3-SAT solution $\implies$ MAXREWARD solution.* Suppose that we have a solution for the 3-SAT problem. It follows that there is an assignment to each variable $v_i$ such that each clause is true. To construct a solution to the MAXREWARD problem, we perform the following. For each variable $v_j$ that is set to true (false) and for each clause $C_l$ containing $v_j$ ($\bar{v}_j$), we play arm $k_j$ ($\bar{k}_j$) at time $m + l$ to obtain a reward of 1, which corresponds to setting $\pi_{m+l}^* = k_j$ ($\pi_{m+l}^* = \bar{k}_j$). Since we can only play an arm each time period, ties can be broken arbitrarily. From this partial solution, we obtain a cumulative reward of $n$ since all of the $n$ clauses are satisfied by at least one literal. Observe that for the remaining arm $\bar{k}_j$ ($k_j$) for the variable $v_j$ that is set to be true (false), it is valid to play $\bar{k}_j$ ($k_j$) at time $j$ to obtain a reward of 1, which corresponds to setting $\pi_j^* = \bar{k}_j$ ($\pi_j^* = k_j$). From this partial solution, we obtain a value of $m$ since there are $m$ variables. It is not hard to see that the constructed $\pi^*$ is a valid solution with $r(\pi^*) \geq n + m = V$.

*MAXREWARD solution $\implies$ 3-SAT solution.* Suppose now that we have a solution for the MAXREWARD problem. It follows that there a solution $i_T^*$ such that $r(\pi^*) \geq V = n + m$. Note that for any feasible solution $\pi$, $r(\pi) \leq n + m$ as from time periods 1 to $m$ and $m + 1$ to $m + n$ we can obtain at most a reward of $m$ and $n$, respectively. Thus, $r(\pi^*) = n + m$. To obtain a reward of $m$ from time period $j = 1, ..., m$, we must play either $k_j$ or $\bar{k}_j$. To obtain a reward of $n$ from time period $j = m + 1, ..., m + n$, we must play exactly one of the arms corresponds to the clause $C_{j-m}$. Thus, to construct an assignment for the 3-SAT instance, we let $v_j$ to be false (true) if $k_j$ ($\bar{k}_j$) is played at time $j$. Such assignment ensures that, for each clause $C_l$, at least one of the literals is true, which corresponds to having one of the (literal) arms played that isn't blocked. $\square$

Together with the above claim, we have completed the reduction and showed that the MAXREWARD problem is strongly NP-hard with bounded path variation. Note that in this proof, we relied on a special set of problem instances where both $T_0$ and $B_{T_0}$ are in the range of max delay $D = n + m$. This might cause some issues for the other analysis in the paper (e.g., the performance guarantee in Theorem 2 is designed for some more well behaved cases). Therefore, we still need to modify the proof above to cover more generic cases.

Now, let $k_1, k_2 \geq 2$ arbitrary integers. For each of the construction above of $T_0$ time steps, we pad it with another $(k_1 - 1)T_0$ time steps where each arm has blocking value 1 and reward value 0. Together with the first $T_0$ time steps, these form an interval of $k_1 T_0$ time steps, called large blocks. We then set $T = k_2 k_1 T_0$ and concatenate $k_2$ copies of these large blocks together. It is easy to see that the proof of reduction above is still valid, but now we have time horizon $T = k_1 k_2 D$, and variation budget $B = k_2 D$. By varying $k_1$ and $k_2$, we can set the relationship of $T$, $B$, and $D$ to be arbitrary. This concludes the proof. $\square$

*Proof of Theorem 2.*  Let $\pi^* = (\pi_1^*, ..., \pi_T^*) \in \arg\max_{\pi \in \mathcal{P}} r(\pi)$ be an optimal solution. Let $\pi^+ = (\pi_1^+, ..., \pi_T^+) \in \mathcal{P}$ be a solution returned by `Greedy-BAA`. Consider a time period $t \in \mathcal{T}$ where

$\pi_t^* \neq \pi_t^+$. There are two cases in which $\pi_t^*$ is not selected by `Greedy-BAA`. The first case is where $X_t^{i_t^*} \leq X_t^{i_t^+}$. The second case is where `Greedy-BAA` played the arm $\pi_t^*$ at (the most recent) time $1 \leq t' < t$ and it is blocked for $D_{t'}^{\pi_{t'}^+}$ time steps. Since $\pi_{t'}^+ = \pi_t^*$, we let $\pi_{t'}^+ = j \in \mathcal{K}$. The difference of the reward is given by

$$|X_{t'}^j - X_t^j| = |X_{t'}^j - X_{t'+1}^j + X_{t'+1}^j - X_t^j| \leq |X_{t'}^j - X_{t'+1}^j| + |X_{t'+1}^j - X_t^j| \leq \sum_{\bar{t}=t'}^{t-1} |X_{\bar{t}}^j - X_{\bar{t}+1}^j|,$$

where the inequalities resulted from adding and subtracting the corresponding terms and applying the triangle inequalities for the absolute value function repeatedly. Thus,

$$X_{t'}^j + \sum_{\bar{t}=t'}^{t-1} |X_{\bar{t}}^j - X_{\bar{t}+1}^j| \geq X_t^j$$

Let $\mathrm{Blk}(t', j)$ be the set of time periods in which playing arm $j$ is optimal in $(\pi_1^*, ..., \pi_T^*)$, but arm $j$ is blocked by playing it at time $t'$ via `Greedy-BAA`. Note that $|\mathrm{Blk}(t', j)| \leq \frac{D_{\max}^j}{D_{\min}^j}$ where where $D_{\max}^j$ and $D_{\min}^j$ are the maximum and minimum blocking duration of arm $j$ across all the time periods, respectively. This is because in the time intervals from $t' + 1$ to $t' + D_{\max}^j$ arm $j$ can be played at most $\frac{D_{\max}^j}{D_{\min}^j}$ times by any algorithm. This gives us

$$\sum_{t \in \mathrm{Blk}(t',j)} X_t^j \leq \sum_{t \in \mathrm{Blk}(t',j)} \left( X_{t'}^j + \sum_{\bar{t}=t'}^{t-1} |X_{\bar{t}}^j - X_{\bar{t}+1}^j| \right) \leq \frac{D_{\max}^j}{D_{\min}^j} \left( X_{t'}^j + \sum_{\bar{t}=t'}^{\max(\mathrm{Blk}(t',j))-1} |X_{\bar{t}}^j - X_{\bar{t}+1}^j| \right).$$

Note that, for any $\bar{t} \neq t'$ such that $\pi_{\bar{t}}^G = \pi_{t'}^G = j$, $\mathrm{Blk}(t', j) \cap \mathrm{Blk}(\bar{t}, j) = \emptyset$.

As a result, each arm $\pi_t^+$ can be used to cover some part of the optimal solution under case 1 and/or case 2 for each time period $t \in \mathcal{T}$. It follows that

$$r(\pi^*) = \sum_{t=1}^T X_t^{\pi_t^*} \leq \sum_{t=1}^T X_t^{\pi_t^+} + \sum_{t=1}^T \frac{D_{\max}^{\pi_t^+}}{D_{\min}^{\pi_t^+}} \left( X_t^{\pi_t^+} + \sum_{\bar{t}=t}^{\max(\mathrm{Blk}(t,\pi_t^+))-1} |X_{\bar{t}}^{\pi_t^+} - X_{\bar{t}+1}^{\pi_t^+}| \right)$$

$$\leq \sum_{t=1}^T X_t^{\pi_t^+} + \frac{D_{\max}^{j^*}}{D_{\min}^{j^*}} \sum_{t=1}^T \left( X_t^{\pi_t^+} + \sum_{\bar{t}=t}^{\max(\mathrm{Blk}(t,\pi_t^+))-1} |X_{\bar{t}}^{\pi_t^+} - X_{\bar{t}+1}^{\pi_t^+}| \right)$$

$$= \sum_{t=1}^T X_t^{\pi_t^+} + \frac{D_{\max}^{j^*}}{D_{\min}^{j^*}} \left( \sum_{t=1}^T X_t^{\pi_t^+} + \sum_{t=1}^T \sum_{\bar{t}=t}^{\max(\mathrm{Blk}(t,\pi_t^+))-1} |X_{\bar{t}}^{\pi_t^+} - X_{\bar{t}+1}^{\pi_t^+}| \right)$$

$$\leq \sum_{t=1}^T X_t^{\pi_t^+} + \frac{D_{\max}^{j^*}}{D_{\min}^{j^*}} \left( \sum_{t=1}^T X_t^{\pi_t^+} + \sum_{i \in \mathcal{K}} \sum_{t=1}^{T-1} |X_t^i - X_{t+1}^i| \right)$$

$$\leq \sum_{t=1}^T X_t^{\pi_t^+} + \frac{D_{\max}^{j^*}}{D_{\min}^{j^*}} \left( \sum_{t=1}^T X_t^{\pi_t^+} + B_T \right) \leq \left( 1 + \frac{D_{\max}^{j^*}}{D_{\min}^{j^*}} \right) r(\pi^+) + \frac{D_{\max}^{j^*}}{D_{\min}^{j^*}} B_T,$$

where the first inequality is from applying case 1 and case 2, the second inequality is from replacing the ratio by $\frac{D_{\max}^{j^*}}{D_{\min}^{j^*}}$, the arm $j^*$ with the highest max-min blocking duration ratio, the third equality by distributing the summations, the fourth inequality by first grouping the time periods that each arm $i$ is played and then applying the sum (which is bounded by $T$), and the fifth inequality is by definition. Rearranging the terms, we obtain our claimed result. □

## B  Proofs for Theorems from Section 4

*Proof of Theorem 3.* Recall that we want to compute the $\alpha$-regret of `RGA` against the optimal offline solution of MAXREWARD. Let $\pi$ denote the policy generated by `RGA`, and recall that $\pi^*$ is the

policy of the optimal solution. For the sake of simplicity, we will refer to $\pi^*$ as $*$ in the indices. Let $\alpha = \frac{D}{D+\tilde{D}}$. The $\alpha$-regret of RGA incurred in a batch $\mathcal{T}_j$ is given by:

$$\sum_{t \in \mathcal{T}_j} (\alpha X_t^* - X_t^\pi) \tag{1}$$

In the first $K$ rounds a loss of at most $K$ can be accumulated. Similarly for the next $\tilde{D}$ time steps and the last $\tilde{D}$ time steps, a loss of at most $\tilde{D}$ is accumulated. Let $\mathcal{T}_j'$ denote the time steps in batch $\mathcal{T}_j$, excluding the first $\tilde{D} + K$ and the last $\tilde{D}$ rounds.

$$\sum_{t \in \mathcal{T}_j} (\alpha X_t^* - X_t^\pi) \leq (2\tilde{D} + K) + \sum_{t \in \mathcal{T}_j'} (\alpha X_t^* - X_t^\pi) \tag{2}$$

Let $\hat{X}_t^\pi$ denote the reward of the arm played by policy $\pi$ at time step $t$ which was observed in the first $K$ rounds of batch $\mathcal{T}_j$ and let $B_j$ denote the path variance within this batch (i.e., batch $\mathcal{T}_j$):

$$B_j = \sum_{t \in \mathcal{T}_j} \sum_{k \in \mathcal{K}} |X_{t+1}^k - X_t^k|$$

Then we have

$$\begin{aligned}
\sum_{t \in \mathcal{T}_j'} (\alpha X_t^* - X_t^\pi) &= \sum_{t \in \mathcal{T}_j'} (\alpha \hat{X}_t^* - \hat{X}_t^\pi) + \sum_{t \in \mathcal{T}_j'} (\alpha X_t^* - \alpha \hat{X}_t^*) + \sum_{t \in \mathcal{T}_j'} (\hat{X}_t^\pi - X_t^\pi) \\
&\leq \sum_{t \in \mathcal{T}_j'} |X_t^* - \hat{X}_t^*| + \sum_{t \in \mathcal{T}_j'} |X_t^\pi - \hat{X}_t^\pi| \\
&\leq \sum_{t \in \mathcal{T}_j'} 2B_j \\
&\leq 2\Delta_T B_j
\end{aligned} \tag{3}$$

The first inequality comes from the fact that $\alpha = \frac{1}{1+\tilde{D}} \leq 1$ and RGA runs Greedy-BAA with fixed estimates $\hat{X}_t$, which is $\alpha$-optimal for an instance of MAXREWARD with fixed $\hat{X}^k$ values (we apply Theorem 2 with variation budget 0). Thus, $\sum_{t \in \mathcal{T}_j'}^T \hat{X}_t^\pi \geq \alpha \sum_{t \in \mathcal{T}_j'}^T \hat{X}_t^*$. The second inequality comes from the following observation: For each $\in \mathcal{T}_j'$ and $k \in \mathcal{K}$, we have

$$|X_t^k - \hat{X}_t^k| \leq \sum_{t, t+1 \in \mathcal{T}_j}^T |X_{t+1}^k - \hat{X}_t^k| \leq B_j \tag{4}$$

Recall that $\hat{X}_t^k$ is the value of first pull of arm $k$ in the batch. Therefore, the difference between that first observed value and $X_t^k$ can be bounded above by the sum of reward changes from round to round of arm $k$, which is further bounded above by the path variation budget $B_j$ of that batch. The third inequality in Eq. (3) comes from the fact that the length of the batch is at most $\Delta_T$. Replacing Eq. (3) into Eq. (2) we get:

$$\sum_{t \in \mathcal{T}_j}^T (X_t^* - X_t^\pi) \leq 2\Delta_T B_j + (2\tilde{D} + K)$$

Summing over all batches we have the following bound on regret:

$$\begin{aligned}
\mathcal{R}_\pi^\alpha(B_T, \tilde{D}, T) &\leq \sum_{j=1}^m 2\Delta_T B_j + \left\lceil \frac{T}{\Delta_T} \right\rceil (2\tilde{D} + K) \\
&\leq 2B_T \Delta_T + \left\lceil \frac{T}{\Delta_T} \right\rceil (2\tilde{D} + K) \\
&\leq 2B_T \Delta_T + \frac{T+1}{\Delta_T} (2\tilde{D} + K)
\end{aligned}$$

Since $B_T \leq TK$ by definition and both $\tilde{D}, K \geq 1$, we have $\sqrt{\frac{(T+1)(2\tilde{D}+K)}{2B_T}} \geq 1$, and thus, $\left\lceil \sqrt{\frac{(T+1)(2\tilde{D}+K)}{2B_T}} \right\rceil \leq \sqrt{\frac{(T+1)(2\tilde{D}+K)}{2B_T}} + 1 \leq 2\sqrt{\frac{(T+1)(2\tilde{D}+K)}{2B_T}}$. By setting $\Delta_T = \left\lceil \sqrt{\frac{(T+1)(2\tilde{D}+K)}{2B_T}} \right\rceil \leq 2\sqrt{\frac{(T+1)(2\tilde{D}+K)}{2B_T}}$ we obtain the desired result.

$\square$

*Proof of Theorem 4.* Let $\pi$ denote META-RGA. The $\alpha$-regret of META-RGA can be expressed as follows:

$$\sum_{i=1}^{\lceil \frac{T}{H} \rceil} \sum_{t=(i-1)H+1}^{\max(T,iH)} (\alpha X_t^* - X_t^\pi)$$

Let $B_i$ denote the total path variance within batch $i$. Of all the RGA instances (i.e., meta-arms) available to there must be an instance who is a associated with a candidate budget $\tilde{B}$ such that:

$$\max_i B_i \leq \tilde{B} \leq 2 \max_i B_i \tag{5}$$

Le $\tilde{\pi}$ denote the policy of this RGA instance. Using $\tilde{\pi}$ we can decompose the regret of META-RGA as follows:

$$\left[ \sum_{i=1}^{\lceil \frac{T}{H} \rceil} \sum_{t=(i-1)H+1}^{\max(T,iH)} (\alpha X_t^* - X^{\tilde{\pi}}) \right] + \left[ \sum_{i=1}^{\lceil \frac{T}{H} \rceil} \left( \sum_{t=(i-1)H+1}^{\max(T,iH)} X_t^{\tilde{\pi}} \right) - \left( \sum_{t=(i-1)H+1}^{\max(T,iH)} X^\pi \right) \right] \tag{6}$$

The second term of Eq (6) can be further bounded as follows. Note that the RGA instance with policy $\tilde{\pi}$ might not be the best fixed meta-arm in hindsight, whose policy is denoted by $\pi^+$. Thus, we have:

$$\left[ \sum_{i=1}^{\lceil \frac{T}{H} \rceil} \left( \sum_{t=(i-1)H+1}^{\max(T,iH)} X_t^{\tilde{\pi}} \right) - \left( \sum_{t=(i-1)H+1}^{\max(T,iH)} X^\pi \right) \right] \leq \left[ \sum_{i=1}^{\lceil \frac{T}{H} \rceil} \left( \sum_{t=(i-1)H+1}^{\max(T,iH)} X_t^{\pi^+} \right) - \left( \sum_{t=(i-1)H+1}^{\max(T,iH)} X^\pi \right) \right]$$

The RHS of this is simply the difference between the rewards observed and accumulated by the Exp3 meta-algorithm and the best available RGA meta-arm in hindsight. Thus we can bound the second term with standard Exp3 regret bounds. Note that there are $\log_2(KT)$ arms available to the Exp3 algorithm, $T/H$ is number of batches, and the maximum reward a meta-arm can receive within a batch is $H$ (i.e., the length of each batch). Thus the second term can be bounded above by $\mathcal{O}\left( H\sqrt{T/H \ln(KT) \ln(\ln(KT))} \right) = \mathcal{O}\left( \sqrt{HT \ln(KT) \ln(\ln(KT))} \right)$.

Now we turn to bound the first term of Eq (6). Each inner sum of the first term correspond to the $\alpha$-regret of policy $\tilde{\pi}$ over a block of length $H$. Our idea is to use Theorem 3 to bound the regret of $\tilde{\pi}$ in each of batches $i$. In order to do so, we must check whether running RGA with budget $\tilde{B}$ in the batches (with time horizon $H$) will result in a valid $\Delta_H$, that is $\Delta_H \geq 1$. From Eq (5) we know that $\tilde{B} \leq 2 \max_i B_i \leq 2HK$ (the second inequality comes from the definition of the total variance budget, which is at most $HK$ for time horizon $H$). Therefore, from Theorem 3 we know that $\Delta_H \geq \sqrt{\frac{(H+1)(2\tilde{D}+K)}{2\tilde{B}}} > \sqrt{\frac{H(2\tilde{D}+K)}{4HK}} \geq 1$ if $\tilde{D} \geq \frac{3K}{2}$. Now, since $\tilde{D}$ is an upper bound of the maximal blocking duration, we can set it to be at least $\frac{3K}{2}$ to make $\Delta_H \geq 1$. Therefore, we can apply Theorem 3 to each of the batches. In particular, the $\alpha$-regret of $\tilde{\pi}$ over batch $i$ of length at most $H$ and with optimally tuned restarts can be bounded as follows:

$$\sum_{t=(i-1)H+1}^{\max(T,iH)} (\alpha X_t^* - X^{\tilde{\pi}}) \leq \sqrt{2\tilde{B}(H+1)(2\tilde{D}+K)}$$

$$\leq 2\sqrt{\tilde{B}H(2\tilde{D}+K)}$$

Summing over all blocks we have:

$$\sum_{i=1}^{\lceil \frac{T}{H} \rceil} \sum_{t=(i-1)H+1}^{\max(T,iH)} (\alpha X_t^* - X^{\tilde{\pi}}) \leq \left( \frac{T}{H} + 1 \right) 2\sqrt{\tilde{B}H(2\tilde{D}+K)}$$

$$\leq 4\frac{T}{\sqrt{H}}\sqrt{\tilde{B}(2\tilde{D}+K)} \tag{7}$$

Combining Eq (7) with the regret bound of the Exp3 meta-bandit algorithm, we get that the $\alpha$-regret of `META-RGA` is at most

$$\mathcal{O}\left( \frac{T}{\sqrt{H}}\sqrt{\tilde{B}(2\tilde{D}+K)} \right) + \mathcal{O}\left( \sqrt{HT\ln(KT)\ln(\ln(KT))} \right) \tag{8}$$

By setting $H = \sqrt{\frac{T(2\tilde{D}+K)}{\ln(KT)\ln(\ln(KT))}}$ we get the desired regret bound. $\qquad\square$

## C Regret Analysis with Other Variation Budgets

In this section we show how our regret analysis can be adopted to the maximum variation budget $B_T^{\max}$ and number of changes budget $L_T$. For the sake of convenience, we repeat the definition of these budgets below:

$$B_T^{\max} = \sum_{t,t+1\in\mathcal{T}} \max_{k\in\mathcal{K}} |X_{t+1}^k - X_t^k|, \quad L_T = \#\{t : 1 \leq t \leq T-1, \exists k : X_t^k \neq X_{t+1}^k\}$$

It is easy to show that $B_T \leq KB_T^{\max} \leq KL_T$. Thus, by just replacing $B_T$ with $KB_T^{\max}$ and $KL_T$ we can already get regret bounds with the other two variation budgets.

However, we show that we can further improve these bounds by order of $\sqrt{K}$ as follows: We only need to modify the way we estimate the regret in Eq (4). In particular, recall that for each $\in \mathcal{T}_j'$ and $k \in \mathcal{K}$, we have

$$|X_t^k - \hat{X}_t^k| \leq \sum_{t,t+1\in\mathcal{T}_j}^{T} |X_{t+1}^k - \hat{X}_t^k| \leq \sum_{t,t+1\in\mathcal{T}_j}^{T} \max_{l\in\mathcal{K}} |X_{t+1}^l - \hat{X}_t^l| \leq B_j^{\max} \tag{9}$$

where $B_j^{\max}$ is the maximum variation budget of batch $j$. Replacing this back to Eq. (3) we get:

$$\sum_{t\in\mathcal{T}_j'} (\alpha X_t^* - X_t^\pi) = \sum_{t\in\mathcal{T}_j'} (\alpha\hat{X}_t^* - \hat{X}_t^\pi) + \sum_{t\in\mathcal{T}_j'} (\alpha X_t^* - \alpha\hat{X}_t^*) + \sum_{t\in\mathcal{T}_j'} (\hat{X}_t^\pi - X_t^\pi)$$

$$\leq \sum_{t\in\mathcal{T}_j'} |X_t^* - \hat{X}_t^*| + \sum_{t\in\mathcal{T}_j'} |X_t^\pi - \hat{X}_t^\pi|$$

$$\leq \sum_{t\in\mathcal{T}_j'} 2B_j^{\max} \tag{10}$$

$$\leq 2\Delta_T B_j^{\max}$$

By following the same steps in the proof of Theorem 3, we get that the regret bound for `RGA` is $\mathcal{O}\left( \sqrt{(2\tilde{D}+K)TB_T^{\max}} \right)$ if $\Delta_T$ is optimally tuned to be $\left\lceil \sqrt{\frac{(T+1)(2\tilde{D}+K)}{2B_T^{\max}}} \right\rceil$.

Similarly, for number of changes $L_T$, we can rewrite Eq. (9) as follows:

$$|X_t^k - \hat{X}_t^k| \leq \sum_{t,t+1\in\mathcal{T}_j}^{T} |X_{t+1}^k - \hat{X}_t^k| \leq \sum_{t,t+1\in\mathcal{T}_j}^{T} \mathcal{I}(X_{t+1}^k \neq \hat{X}_t^k)$$

$$\leq \sum_{t,t+1\in\mathcal{T}_j}^{T} \mathcal{I}(\exists l \in \mathcal{K} : X_{t+1}^l \neq \hat{X}_t^l) \tag{11}$$

$$\leq L_j$$

where $\mathcal{I}(\cdot)$ is the indicator function, and $L_j$ is the total number of changes in batch $j$. The rest is similar to the discussion above, and we can get $\mathcal{O}\left(\sqrt{(2\tilde{D}+K)TL_T}\right)$ regret bound for our algorithm (with $\Delta_T = \left\lceil \sqrt{\frac{(T+1)(2\tilde{D}+K)}{2L_T}} \right\rceil$).

Regarding the new values for the approximation ratio of `Greedy-BAA`, recall that the approximation bound for `Greedy-BAA` can be calculated as follows:

$$r(\pi^*) = \sum_{t=1}^{T} X_t^{\pi_t^*} \leq \sum_{t=1}^{T} X_t^{\pi_t^+} + \sum_{t=1}^{T} \frac{D_{\max}^{\pi_t^+}}{D_{\min}^{\pi_t^+}} \left( X_t^{\pi_t^+} + \sum_{\bar{t}=t}^{\max(\mathrm{Blk}(t,\pi_t^+))-1} |X_{\bar{t}}^{\pi_t^+} - X_{\bar{t}+1}^{\pi_t^+}| \right)$$

$$\leq \sum_{t=1}^{T} X_t^{\pi_t^+} + \frac{D_{\max}^{j^*}}{D_{\min}^{j^*}} \left( \sum_{t=1}^{T} X_t^{\pi_t^+} + \sum_{i \in \mathcal{K}} \sum_{t=1}^{T-1} |X_t^i - X_{t+1}^i| \right) \tag{12}$$

Note that the term $\sum_{i \in \mathcal{K}} \sum_{t=1}^{T-1} |X_t^i - X_{t+1}^i|$ from Eq (12) can be bounded above by $KB_T^{\max}$ and $KL_T$, respectively. This implies that:

$$r(\pi^*) = \sum_{t=1}^{T} X_t^{\pi_t^*} \leq \sum_{t=1}^{T} X_t^{\pi_t^+} + \frac{D_{\max}^{j^*}}{D_{\min}^{j^*}} \left( \sum_{t=1}^{T} X_t^{\pi_t^+} + \sum_{i \in \mathcal{K}} \sum_{t=1}^{T-1} |X_t^i - X_{t+1}^i| \right)$$

$$\leq \sum_{t=1}^{T} X_t^{\pi_t^+} + \frac{D_{\max}^{j^*}}{D_{\min}^{j^*}} \left( \sum_{t=1}^{T} X_t^{\pi_t^+} + KB_T^{\max} \right)$$

$$\leq \left( 1 + \frac{D_{\max}^{j^*}}{D_{\min}^{j^*}} \right) r(\pi^+) + \frac{D_{\max}^{j^*}}{D_{\min}^{j^*}} KB_T^{\max} \tag{13}$$

Similarly, we have:

$$r(\pi^*) = \sum_{t=1}^{T} X_t^{\pi_t^*} \leq \sum_{t=1}^{T} X_t^{\pi_t^+} + \frac{D_{\max}^{j^*}}{D_{\min}^{j^*}} \left( \sum_{t=1}^{T} X_t^{\pi_t^+} + \sum_{i \in \mathcal{K}} \sum_{t=1}^{T-1} |X_t^i - X_{t+1}^i| \right)$$

$$\leq \sum_{t=1}^{T} X_t^{\pi_t^+} + \frac{D_{\max}^{j^*}}{D_{\min}^{j^*}} \left( \sum_{t=1}^{T} X_t^{\pi_t^+} + KL_T \right)$$

$$\leq \left( 1 + \frac{D_{\max}^{j^*}}{D_{\min}^{j^*}} \right) r(\pi^+) + \frac{D_{\max}^{j^*}}{D_{\min}^{j^*}} KL_T \tag{14}$$

Thus, we have the approximation ratio of $\left[ \left(1 + \tilde{D}\right) + \tilde{D}KB_T^{\max}/r(\pi^+) \right]^{-1}$ for the usage of maximum variation budget $B_T^{\max}$, and $\left[ \left(1 + \tilde{D}\right) + \tilde{D}KL_T/r(\pi^+) \right]^{-1}$ if we use the number of changes budget $L_T$.

# D  Proof of Claims 1 and 2

*Proof of Claim 1.* Consider the case of $B_T = KT$. This implies that the rewards can change in an arbitrary way. Now consider the case when $D_t^k = 1$ for all $k \in \mathcal{K}$ and $t \in \mathcal{T}$ (i.e., there is no blocking at all). In this case, we have $\alpha = 1/2$. The main idea of the proof is to randomly generate the sequences of $X_t^k$ in some way and prove that in expectation (over this randomisation), the $\alpha$-regret is large. In particular, for any arm pulling policy $\pi$ we have that:

$$\mathbb{E}\left[ \alpha \sum_t X_t^* - \sum_t X_t^\pi \right] = \mathbb{E}\left[ \alpha \sum_t \max_{k \in \mathcal{K}} X_t^k - \sum_t X_t^\pi \right]$$

$$\geq \mathbb{E}\left[ \alpha \sum_t \max_{k \in \mathcal{K}} X_t^k \right] - \max_{k \in \mathcal{K}} \mathbb{E}\left[ \sum_t X_t^k \right] + \max_{k \in \mathcal{K}} \mathbb{E}\left[ \sum_t X_t^k \right] - \mathbb{E}\left[ \sum_t X_t^\pi \right]$$

$$\geq \alpha \sum_t \mathbb{E}\left[ \max_{k \in \mathcal{K}} X_t^k \right] - \max_{k \in \mathcal{K}} \mathbb{E}\left[ \sum_t X_t^k \right] + \tilde{R}_T^\pi$$

$$\tag{15}$$

where $\tilde{R}_T^\pi$ is the pseudo regret of $\pi$ against the best fixed policy in hindsight. Now we use the standard stochastic setup to prove the lower bound of the pseudo regret: e.g., the arms are drawn from Bernoulli distributions with one arm to have reward mean of $\varepsilon + \sqrt{\frac{K}{T}}$, while the other arms have reward mean of $\varepsilon$ (see, e.g., [?] for the technical details). By doing so, we can prove that $\tilde{R}_T^\pi \geq \frac{1}{8}\sqrt{KT}$. In addition, we have that:

$$
\begin{aligned}
\alpha \sum_t \mathbb{E}\Big[ \max_{k \in \mathcal{K}} X_t^k \Big] &- \max_{k \in \mathcal{K}} \mathbb{E}\Big[ \sum_t X_t^k \Big] \\
&= \alpha T\Big(1 - (1-\varepsilon)^K(1 - \sqrt{K/T} - \varepsilon)\Big) - T(\sqrt{K/T} + \varepsilon) \qquad (16) \\
&= T\Big(\alpha\Big(1 - (1-\varepsilon)^K(1 - \sqrt{K/T} - \varepsilon)\Big) - \Big(\sqrt{K/T} + \varepsilon\Big)\Big)
\end{aligned}
$$

Substituting $\alpha = 1/2$ and $\beta = (1-\varepsilon)^K$ we further have:

$$
\begin{aligned}
T\Big(\alpha\Big(1 - (1-\varepsilon)^K(1 - \sqrt{K/T} - \varepsilon)\Big) &- \Big(\sqrt{K/T} + \varepsilon\Big)\Big) \\
&= T\Big((1-\beta)/2 - (\sqrt{K/T} + \varepsilon)(1 - \beta/2)\Big) \qquad (17) \\
&\geq T\Big((1-\beta)/2 - \varepsilon\Big) - \sqrt{KT}
\end{aligned}
$$

Putting all these together we get:

$$
\mathbb{E}\Big[\alpha \sum_t X_t^* - \sum_t X_t^\pi\Big] \geq T\Big((1-\beta)/2 - \varepsilon\Big) - \frac{7}{8}\sqrt{KT} \qquad (18)
$$

It is easy to show that for any $K \geq 2$, with a sufficiently small $\varepsilon$, there exists a constant $c > 0$ such that $(1-\beta)/2 - \varepsilon > c$. This implies that $\mathbb{E}\Big[\alpha \sum_t X_t^* - \sum_t X_t^\pi\Big] \in \Theta(T)$, which concludes the proof. $\qquad \square$

Note that the same proof works for any constant $\alpha > 0$.

*Proof of Claim 2.* Note that in this claim, the value of $\alpha$ is defined by the approximation ratio of `Greedy-BAA`, and not the value from Theorem 3. The reason for this modification is that $\alpha = \frac{1}{1+\tilde{D}}$ in this case becomes $\Theta(1/T)$, which is not very meaningful. In particular, $\alpha = 1/T$ implies that, as the optimal solution is bounded above by $T$, a policy with sub-linear $\frac{1}{T}$-regret would only need to achieve $\Theta(1)$ performance, which is not difficult to achieve.

Now, consider the following two instances of a 2-arm bandit model: In problem instance P1, we have a bandit model with arms 1 and 2. For $t = 1$, we have $X_1^1 = 1$ with $D_1^1 = 1$, and $X_1^2 = 0$ with $D_1^2 = T$, respectively. For $t \geq 2$ we set $X_t^1 = 0$ with $D_t^1 = 1$, and $X_t^2 = 1$ with $D_t^2 = 1$ as well. It is clear that in this instance `Greedy-BAA` will also be the optimal solution, with pulling Arm 1 at $t = 1$ and repeatedly pulling Arm 2 afterwards (thus, the optimal performance is $T$). If any policy starts with pulling Arm 2 first, the total reward it can collect is 0 (as after pulling Arm 2 at $t = 1$, from $t = 2$, the only feasible arm is Arm 1 with reward 0).

We also design problem instance P2 by swapping the rewards and blocking durations of the 2 arms in P1 with each other. In this instance, the optimal solution is to pull Arm 2 first and then repeatedly pull Arm 1. For both P1 and P2, the path variation budget is $B = 2$.

Now, consider an arbitrary policy $\pi$. Suppose that $\pi$ pulls Arm 1 with probability $p \in [0,1]$. For now, assume that $p \leq 1/2$. In this case, if $\pi$ is applied to P1, its expected reward will be $pT + (1-p)0 \leq T/2$, implying that the difference between the performance of $\pi$ and that of `Greedy-BAA` is at least $T/2$. Similarly, if $p > 1/2$, we will consider P2. Putting these together we can see that for any arbitrary policy $\pi$, there exists a problem instance on which the approximate regret is at least $T/2$. $\qquad \square$

# E    Performance Comparison between Greedy-BAA and RGA

From Theorem 2 we have that:

$$r(\pi^*) \le \left(1 + \frac{D_{\max}^{k^*}}{D_{\min}^{k^*}}\right) r(\pi^+) + \frac{D_{\max}^{k^*}}{D_{\min}^{k^*}} B_T \le \left(1 + \frac{\tilde{D}}{\underset{\sim}{D}}\right) r(\pi^+) + \frac{\tilde{D}}{\underset{\sim}{D}} B_T$$

where $r(\pi^*)$ is the (offline) optimal solution, and $r(\pi^+)$ is the performance of `Greedy-BAA`. This can be rewritten as:

$$\frac{\underset{\sim}{D}}{\underset{\sim}{D} + \tilde{D}} r(\pi^*) - \frac{\tilde{D}}{\underset{\sim}{D} + \tilde{D}} B_T \le r(\pi^+). \tag{19}$$

For `RGA`, we know from Theorem 3 that

$$\frac{\underset{\sim}{D}}{\underset{\sim}{D} + \tilde{D}} r(\pi^*) - \mathcal{O}\left(\sqrt{T(2\tilde{D} + K)B_T}\right) \le r(\text{RGA}). \tag{20}$$

If $B_T = o(T)$, we have that $\sqrt{T(2\tilde{D} + K)B_T} > B_T$, thus the LHS of Eq (19) is larger than the LHS of Eq (20), which implies that the approximation ratio of `Greedy-BAA` is still a better performance guarantee than that of `RGA` (i.e., the the $\alpha$-regret bound).

We further demonstrate this by running a small numerical experiment as follows: In this experiment we set $T = 10000$, $K = 10$, and the initial maximal path variation $B_T = 3$ (our results show a similar broad view for other parameter settings as well). We compare the performance of Greedy-BAA, RGA, and a random algorithm (which uniformly and randomly pull a feasible arm at each time step).

We consider reward vectors that have a reward of $1$ for one arm and $0$ for the others. We then divide the time horizon into switching blocks of fixed length. In each switching block the reward vector switches to another reward vector uniformly at random. For each fixed switching block size, We run the experiment $50$ times, and plot the average performance in Figure 1 (the error bars with confidence value of $0.95$ are too small that they were removed from the plots for the sake of visualisation). In particular, we plot the average collected reward value against the switching block size.

Note that for both the uniform random and `Greedy-BAA` algorithms the average reward is constant regardless of the size of the block chosen. This makes intuitive sense as `Greedy-BAA` sees the reward vector ahead of time so knows when to switch arms, whilst the random algorithm is just pulling arms at random and makes no attempt to track the best arm. Note that making the switching blocks large corresponds to reducing the variation budget. This is interesting as it seems, that as switching block length increases, `RGA` begins to approach the performance of `Greedy-BAA`. When the switching block size becomes too small the performance of `RGA` deteriorates and becomes equivalent to the performance of the uniform random policy. But in all the cases, the average performance of `RGA` is still below that of `Greedy-BAA`.

Figure 1: A performance comparison between Greedy-BAA (red), RGA (blue), and uniform random (green).