[Reviews · NeurIPS 2020]

Review 1

Summary and Contributions: The authors consider a multi-armed bandit problem with arm blocking (after each play) where delays and rewards are adversarial. They prove in the offline version, i.e. when the delays and rewards are known, computing the reward maximizing arm pull sequence is NP-hard. In the offline, setting they show the Greedy algorithm that plays the best available arm in each time achieves a non-trivial approximation ratio as function of various parameters of the system. In the online version, they consider the situation when the delays and rewards are unknown; but the path variance (B_T), maximum delay (D_up), number of arms (K), and time horizon (T) are known. In this setting, they design a repeating greedy algorithm (RGA) which is divided into a specific number of phases. In each phase, the algorithm samples all the arms once first. Then using these samples as the mean value plays greedily for the rest of the phase. This provides an alpha regret of O(sqrt((D_up+K) T B_T)) with alpha = O(1/D_up). Furthermore, when the path variance B_T is unknown they provide an EXP3 based meta-algorithm (meta-RGA) that provides an O(T^3/4) alpha regret for alpha = O(1/D_up).

Strengths: The author provides an interesting extension of the blocking bandits problem with adversarial delay and rewards. In the offline setting the hardness and approximation results seem to be novel. In the bandit setting, the algorithms is novel as it does not maintain a probability distribution over the arms. It simulates Greedy offline algorithm with means of the arms sampled frequently enough. The meta-algorithm eliminates the need for the knowledge of path variance (which is impractical). However, it suffers a larger regret.

Weaknesses: For the offline problem, I feel, the references are not adequate. The authors should more carefully look for similar setting in the literature of scheduling algorithms. Further, the authors should describe the relation with the adversarial bandits with budget constraints [1,4,5] (citations in the paper) more clearly (e.g. local vs global constraints). Edit: The response is mostly satisfactory. Mention the approximation ratio of the other related scheduling papers, if possible. The presence of the reward obtained by the Greedy algorithm in the offline approximation guarantee is not desirable. The authors should try and remove this dependence. Using some trivial lower bound on the Greedy algorithm is one way to proceed. Edit: Not entirely satisfactory, as without removing the dependence on $r(\pi^+)$ it is very hard to evaluate the results in a sense that is standard in literature. In fact, I think, the following more transparent ratio is achievable (please verify): (1 - \frac{\tildeD}{\utildeD} \frac{\tildeD B_T}{\mu^* T} ) / (1 + \frac{\tildeD}{\utildeD}) When the maximum and the minimum delay ratio is not big, and the path variance is small, then in the bandit version approximation guarantee is much weaker than the offline version. Is this unavoidable? Why does the analysis becomes loose? Edit: The response is not satisfactory as my comment was on the approximation ratio, not the regret guarantee in the online setting. The example in the response, still preserves the same approximation ratio in the online case. When B_T is small (e.g. O(1), log(T)), the phase length Delta_T is long. The authors should discuss why not keeping track of the rewards during each phase is a good idea. They may add some discussion if some other approach is desirable in this regime. Edit: The response clarifies my doubts for the above point.

Correctness: I have checked the proof to be logically correct. I have not looked into the algebraic calculations presented for various inequalities.

Clarity: The paper is clearly written.

Relation to Prior Work: The relation to the prior work in the online learning literature is described well. Reviewing of literature in the scheduling algorithms literature to establish the novelty of the results in the offline setting will improve the placement.

Reproducibility: Yes

Additional Feedback:


Review 2

Summary and Contributions: The paper considers a bandit setting where pulling an arm makes the arm unavailable for a number of rounds. The setting has adversarial rewards that satisfy a condition called “bounded path variation” which limits how much the reward of an arm can change from timestep to timestep (in aggregate). The paper considers 3 feedback models: everything being known in advance, the rewards and block durations being known at the start of a round, and the reward and block duration of the pulled arm only being revealed after pulling the arm. The authors show that the offline problem (first feedback model) is NP-complete when blocking periods are O(T). This motivates looking for approximation algorithms for the other feedback models. For the second feedback model, the authors present a simple greedy algorithm that linear in the ratio of largest to smallest blocking length. Finally the authors present 2 algorithms for the final feedback model (one for known path variation, and one for unknown path variation) and they show these are vanishing-alpha-regret algorithms with alpha linear in block duration and the remaining regret vanishing as sqrt.

Strengths: 1. The adversarial reward model (with bounded path variation) seems like a nice model to prove results for, and I appreciate that the authors point out in section 5 how results from this model translate to related models like “maximum variation budget”. 2. The authors are comprehensive in their treatment of feedback models and present solutions for all of them.

Weaknesses: 1. [See Additional Feedback for post-rebuttal comments] The hardness result for the offline problem relies on blocking arms for the entire remaining duration. Presumably this isn’t the typical type of MAXREWARD instance that we’re primarily interested in (especially since the algorithms presented also break down in this case, see e.g. line 282). It’d be nice if we could say something about the case where blocking periods are bounded, e.g. by O(sqrt(T)). 2. For Thm 2 (and section 4), it seems this bound could be quite large (i.e. O(max blocking length)). It would be nice to show whether this is necessary (i.e. is there a lower bound), or if this is a function of the chosen algorithm.

Correctness: The claims seem correct

Clarity: Generally the paper is well-written

Relation to Prior Work: yes

Reproducibility: Yes

Additional Feedback: ### POST-REBUTTAL I find the hardness instance given in the rebuttal to still be somewhat unsatisfactory. For Thm 1, the authors (in the submission) need to use blocking length D equal to T for some arms, and 1 for others. If we look at the theorem statement for Thm 2, the bound is only non-vacuous if D_max / D_min is bounded by a reasonable function of T, but at least on the hardness result instance, the bound D_max = T while D_min = 1, so the approximation ratio is larger than the max reward that could be earned (since rewards are in [0,1]. The authors proposed solution in the rebuttal pads the instance with 0 rewards, which doesn't seem to resolve the vacuousness of the statement since D_max / D_min is still the same as the total reward that could be achieved. In the reviewer discussion a separate candidate instance for the hardness result was proposed where the greedy algorithm gets a non-trivial guarantee: Use the original instance for (n+m) rounds then 0-pad for (n+m) rounds, and then repeat these 2(n+m) round structure enough times to get to total T time steps. I encourage the authors to validate that the instance indeed proves hardness, and that the greedy algorithm has a non-trivial performance on this instance, and trusting that the authors will do this I have raised my score to a 6.


Review 3

Summary and Contributions: The paper addresses the problem of adversarial blocking bandits when the rewards and unavailability of the arms are not generated by a stochastic process (they are fixed before the decision maker begins playing). This problem is relevant in practice as many online resource allocation settings fit into this mold. The authors further endow the setting with a so called "path variation budget" - a cumulative limit on how much the reward of any arm can vary over time. In this setting 3 main results are presented: 1) Showing the problem of computing the optimal sequence of arms is strongly NP-hard - meaning it is intractable, in practice, to compute the optimal sequence of plays even with full information on the sequence of rewards and delays of all arms. 2) A proxy optimal policy is proposed for this setting Greedy-BAA which plays greedily but has with full knowledge of the rewards and delays at each individual time-step, but no further. For this policy, the authors are able to recover and approximation ration relative to the best possible policy. While weaker than the state-of-the-art results for settings where the reward do not vary over time, this approximation ration manages to be more general and account for the proposed path variation budget. 3) Finally, the authors propose the RGA and the Meta-RGA algorithm, the latter not requiring knowledge of the variation budget and provide bounds on their (\alpha-)regret.

Strengths: I think the results presented in this paper are sound and the problem is interesting. The contribution seems novel and provides new insights into the blocking bandit problem when blocking durations vary over time and rewards exhibit bounded cumulative change over the entire length of the game. The significance (and relevance in the practical world) of the problem is well justified in the introduction. I am pleased to find the numerical experiment in the appendix but I would have liked to see a more comprehensive numerical analysis of the algorithm. I have not checked the correctness of the proofs in the appendix.

Weaknesses: I think the practical relevance of the paper would have been more evident if there was a more comprehensive numerical analysis with more problem settings being studied. More discussion would have also been useful, particularly over the relationship between the problem settings in the experiment relative to the space of parameters (which ones are harder and why? what characteristic or mechanism of the algorithm is highlighted in each etc.). A "path variation budget" of 3 seems very low. Edit after author feedback: I have read and am satisfied with the authors' reply. I will maintain my score and vote for acceptance.

Correctness: I have not checked the proofs in the appendix. The experiment in the appendix is not comprehensive enough to allow the drawing of meaningful conclusions, in my opinion.

Clarity: For the most part the paper is well written however the results are a bit hard to parse. The presentation could be improved by making some of the mathematical statements more self-contained: - in the problem description, it would help if it were clearly mentioned if there are any restrictions on how long arms can remain unplayable (what is the domain of D_min and D_max). Similarly, in the problem description mention the settings that are being addressed in the paper ( are D and X known? the paper handles 3 cases, when the entire sequences are known, the rewards of only the next rounds are visible and the bandit setting). Having this mentioned clearly in the problem setting removes a lot of confusion for the reader early on. - on line 151 What is \bb{U}? - In Claim 1, please be explicit as to what \pi is. - In the proof of theorem 1 could need expansion and clarifying. I could not easily follow the arguments presented inside. Typo: line 205: Much more complicated

Relation to Prior Work: I believe this paper's relation to previous contributions is clear.

Reproducibility: Yes

Additional Feedback:


Review 4

Summary and Contributions: The blocking bandits model was introduced in a paper by Basu et al [Neurips'19] for the stochastic setting. This paper studies the same problem in the adversarial setting. Blocking bandits is a MAB setting where an arm is blocked for some rounds after it is pulled (for example, if we schedule a job to a server, the server will be unavailable for some time while it completes the job). The contributions are as follows: (i) offline: first the authors ignore the learning aspect of the problem and treat it as an offline optimization problem where rewards and blocking times are known ahead of time. In this setting they show that the problem is strongly NP hard (ii) online: the they treat it as an online algorithms problem and propose an approximation by a greedy policy. This is equivalent to a learning setting with full feedback (iii) bandit feedback: finally with bandit feedback (where they only learn the rewards and blocking times for the arm pulled) they give a reduction to the online algorithm that works as follows. The time horizon is divided into epochs and in each epoch, a bunch of arms are pulled to get samples of the rewards and blocking times and then those are used in an instance of the Greedy Algorithm. The algorithm needs to know a bound on the path length variation. (iv) if the bound of the path length variation variation is not known then an Exp3 algorithm is added on top to select the best bound.

Strengths: * the setting is interesting and models well several nice application. * the authors are very thorough in the sense that they explore all natural variations about the problem

Weaknesses: * for the offline and online part, I miss a discussion on the relation to scheduling. There is a vast literature on solving allocation problems with blocking constraints (see this excellent book https://arxiv.org/pdf/2001.06005.pdf for example). Many of those variations are known to be NP-hard and have classical approximation algorithms. While i didn't try myself to map it to the correct problem, I feel the authors should at least compare to the main problems in scheduling and argue why it is different (if it is). My hunch is that it may be a special case of one of those problems. * I think the learning angle (and bandit feedback) is new and interesting, but I am somewhat underwhelmed by the actual algorithm. It seems like the main idea follows the standard reduction form bandit to full feedback (with some non-trivial adaptation, but the main idea still seems like the standard reduction).

Correctness: The results appear to be correct, but I didn't check all the details.

Clarity: Yes.

Relation to Prior Work: I think a comparison with work on the scheduling literature is missing and it would be very important for this paper.

Reproducibility: Yes

Additional Feedback: My main concern is that there should be more discussion on the relation with scheduling. I am also concerned that the learning part follows from a somewhat standard reduction to the full information setting.

[Author Response · NeurIPS 2020]

We would like to thank the reviewers for their valuable comments. We first address the common criticisms, then turn to
each specific comments in what follows.

**Missing relevant work from the scheduling literature**: A common criticism from multiple reviewers is the lack of
mentioning about the relationship of MAXREWARD with scheduling problems in the paper. Indeed, there is a strong
connection between MAXREWARD and the interval scheduling problems. We have removed the description of this
connection from the submitted version mainly due to lack of space ( also, we decided to prefer the other related work
and thus kept them instead in the paper to comply with the previous blocking bandit papers). We sincerely apologise for
this mistake and will add it back to our paper in the next version. This connection is described below in more detail:

The MAXREWARD problem belongs to the class of fixed interval scheduling problems with arbitrary weight values,
no preemption, and machine dependent processing time (see e.g., Kolen *et al.* 2007 for a comprehensive survey). This
is one of the most general, and thus, hardest versions of the fixed interval scheduling literature (see, e.g., Kovalyov,
Ng & Cheng 2007 for more details). In particular, MAXREWARD is a special case of this setting where for each
task, the starting point of the feasible processing interval is equal to the arrival time. Note that to date, provable
performance guarantees for fixed interval scheduling problems with arbitrary weight values only exist in offline, online
but preemptive, or settings with some special uniformity assumptions (Erleback & Spieksma 2000, Miyazawa &
Erleback 2004, Bender *et al.* 2017, Yu & Jacobson 2020). Therefore, to our best knowledge, *Theorem 2 in our paper is*
*the first result which provides provable approximation ratio for a deterministic algorithm* in an online non-preemptive
setting. Note that with some modifications, *our proof can also be extended to the general online non-preemptive setting*,
i.e., online interval scheduling with arbitrary weight values, no preemption, and machine dependent processing time.

**R1**. *Re: the presence of the reward of the Greedy algorithm in the approximation guarantee is not desirable:* Indeed, we
can remove the dependence on the performance of the online greedy algorithm in the approximation ratio as suggested
by the reviewer. For example, when $D \in O(1)$, we have $r(\pi^+) \in \Omega(T)$. Therefore, for settings with $B_T = o(T)$ we
get constant approximation ratio. Note that we also mentioned this in line 208. The reason we still used the form
described in Theorem 2 is to provide a convenient way to compare the performance of the online greedy with the
proposed bandit algorithm (see Appendix E for more details). We will update our paper to reflect this comment.

*Re: the bandit version's approximation guarantee is much weaker than the offline version when the delays are not big,*
*and the path variance is small:* This is indeed unavoidable. For example, consider the case of $D = 1$ for all the arms
and time steps (i.e., there is no blocking). In this scenario, it is easy to see that online greedy becomes optimal. On the
other hand, it is also known that in this case, the regret lower bound of bandit algorithms (against the optimal solution)
is $\Theta(\sqrt{TB})$ (see, e.g., Auer et al. 2002, Cesa-Bianchi & Lugosi 2006, Lattimore & Szepesvári 2019).

*Re: keeping track of the rewards during each phase when $B_T$ is small and the phase length $\Delta_T$ is long:* This is indeed
a good idea, as when $D = 1$ (i.e., there is no blocking), Optimistic Mirror Descent (OMD) works with this insight and
typically gives the best $B_T$ dependent bound (Wei & Luo 2018). However, OMD requires maintaining a probability
distribution over all the arms and this is not possible in our setting because of arbitrary blockings. $B_T$ measures the
change in reward over consecutive rewards, but tracking such a change is not possible in a round if an arm is blocked.

**R2.** *Re: The hardness result for the offline problem with small blocking values:* We can easily extend our current proof
to the case of $T >> D$. In particular, Let $T_0 = n + m = D$. We use the same proof in the paper but replace $T$ with $T_0$.
Now assume that $T >> T_0$ (and thus, $T >> D$). For any $T_0 < t \leq T$, we set the rewards to be 0 and blocks = 1 for
all the arms. It is still true that the optimal solution of this instance is linked to the solution of the original 3-SAT.

*Re: Whether the O(max blocking length) performance gap is necessary:* We would like to highlight that there are 2
performance gap results in our paper: (i) The approximation ratio between the online greedy and that of the offline
optimal, and (ii) the regret between the bandit setting and the online greedy algorithm. For the latter, after the submission
of the paper we have managed to derive a general lower bound of $\Theta(\sqrt{BDT})$ (to prove this we reduced the problem of
combinatorial bandits with limited changes to our setting). Thus, the dependence of the regret bound on $\Theta(\sqrt{D})$ is
necessary. For the approximation ratio of the online greedy, it is true that we do not know whether our result is tight.
Therefore, it remains future work to investigate this case.

**R3.** *Re: more comprehensive numerical analysis needed:* We indeed only focus on the theoretical analysis of the
blocking bandit model. The numerical results in Appendix E is only for supporting the theoretical comparison between
Greedy-BAA and RGA. In particular, Eqs (19) and (20) from Appendix E show that Greedy-BAA is significantly better
than RGA when $B_T$ is small (i.e., the regret bound of RGA is $O(\sqrt{T/B_T})$-time larger). Hence the choice of $B_T = 3$.

**R4.** *Re: It seems like the main idea follows the standard reduction form bandit to full feedback with some non-trivial*
*adaptation:* We agree with the reviewer that the theoretical analysis of the bandit part is a non-trivial adaptation of
known techniques. However, we believe that this part still has its merits, as it provides a neat analysis for a new
and interesting bandit problem, laying the foundation for other adversarial blocking bandit models (e.g., contextual,
combinatorial, etc). This, combined with the other contributions of the paper, make our findings novel.

[Meta-Review · NeurIPS 2020]

The reviewers mostly agree that the paper makes valuable contributions to studying learning in blocking bandits without purely stochastic assumptions, and systematically investigates the hardness of both planning and learning depending on the information structure available to the learner (advance information vs. online (bandit)). The reviewers engaged in a detailed discussion after the author feedback was received, in which several illuminating observations and suggestions were brought up. In view of the positive signals received from the reviewers, I recommend acceptance. I would request that the author(s) pay close attention to the additional feedback from the reviewers and incorporate the suggestions when preparing the final version, especially those from R1 and R2 whose comments were quite insightful.